# Socioeconomic inequalities associated with Geriatric syndrome in Thailand: The results of Fifth National Health Examination Survey

Supakorn Sripaew[1,2], Sawitri Assanangkornchai[2]*, Jiraluck Nontarak[3], Suwat Chariyalertsak[4], Pattapong Kessomboon[5], Surasak Taneepanichskul[6], Nareemarn Neelapaichit[7], Wichai Aekplakorn[8]

1 Department of Family and Preventive Medicine, Faculty of Medicine, Prince of Songkla University, Hat Yai, Songkhla, Thailand, 2 Department of Epidemiology, Faculty of Medicine, Prince of Songkla University, Hat Yai, Songkhla, Thailand, 3 Department of Epidemiology, Faculty of Public Health, Mahidol University, Bangkok, Thailand, 4 Faculty of Public Health, Chiang Mai University, Chiang Mai, Thailand, 5 Faculty of Medicine, Khon Kaen University, Khon Kaen, Thailand, 6 College of Public Health Sciences, Chulalongkorn University, Bangkok, Thailand, 7 Ramathibodi School of Nursing, Faculty of Medicine, Ramathibodi Hospital, Mahidol University, Bangkok, Thailand, 8 Department of Community Medicine, Ramathibodi Hospital Mahidol University, Bangkok, Thailand

* savitree.a@psu.ac.th

**Data Availability Statement:** The data that support the findings of this study are available from the National Health Examination Survey Office, Health

## Abstract

Geriatric syndrome (GS) is the prevalence of a group of phenotypes in older people. Functional decline, cognitive impairment, and frailty are common phenotypes that burden individuals, families, and the healthcare system. Policies targeting GS require information on socioeconomic background of older people, which is scarce in Thailand. We investigated socioeconomic inequality associated with GS using the concentration index and further explained the contributions of socioeconomic status and sociodemographic variables to inequality. Nationally representative data of 7,365 individuals aged 60 years and above from the 5th National Health Examination Survey of 2013 were analyzed. The survey used a physical examination, blood test, and questionnaire interviews to elicit personal information, health status, and household assets. The wealth index was used as the main indicator of socioeconomic status, and participants with missing wealth index data were excluded. Three GS phenotypes—frailty, functional impairment (FI) and neurocognitive dysfunction (NCD)—were included. An indirectly standardized concentration index ($C^{is}$) and a 95% confidence interval were used to represent the horizontal equity of the three phenotypes. Contributions to the concentration index (CC)—contribution to a more or less equitable GS distribution—were decomposed and shown in terms of percentage and direction. All GS phenotypes were found to be concentrated in the elderly poor ($C^{is}$ of FI, frailty, and NCD = -0.068, -0.092, and -0.182, respectively). Work status contributes to a more equitable GS distribution in all the phenotypes (%CC in FI, frailty, and NCD = -1.7%, -5.1%, and -2.0%, respectively), whereas types of insurance schemes made bidirectional contributions to the equity of GS. Policies should be adopted to help prevent GS among poor individuals, provide them with an equal opportunity of access to health schemes and ensure opportunities for older Thai individuals to work.

Systems Research Institute, Thailand. However, we cannot make publicly available the data we used for the current analysis as the investigator committee of the Fifth National Health Examination Survey owns it. The restriction applied to data sharing could be lifted and the data could be made available upon reasonable request (Contact: Department of Community Medicine, Faculty of Medicine, Ramathibodi Hospital, Bangkok, Thailand, email: saraban-ra-ramahosp@mahidol.ac.th).

**Funding:** Funding The fifth National Health Examination Survey (NHES-5) was supported by the Bureau of Policy and Strategy, Ministry of Public Health under the grant number 57–054, Thai Health Promotion Foundation (grant number 55-00-1177), the National Health Security Office, Thailand and the National Research Council of Thailand (127/2564). The NHES-5 was conducted by the National Health Examination Survey Office, Health Systems Research Institute, Thailand. The funders did not play any role in study design, data collection and analysis, decision to publish, or preparation of the manuscript. Supakorn Sripaew received funding support for the publication of this article from the Faculty of Medicine, Prince of Songkla University.

**Competing interests:** The authors have declared that no competing interests exist.

## Introduction

Aging has great socioeconomic impact in several countries [1]. Globally, older adults have been progressively increasing for the past decades; such an increase has greatly affected population structure which in turn affects overall dependency rate and potentially creates socioeconomic insecurity [1, 2]. Further, burden from diseases also shifts towards chronic non-communicable diseases and aging conditions [3, 4]. Geriatric syndrome (GS) is a wide spectrum of conditions that are prevalent in older people and critically impact elderly health in terms of reducing functionality and increasing overall mortality [5]. Several studies reported that older adults who had GS experienced higher all-cause mortality, [6] disability, and limiting quality of life [5, 7]. The syndrome covers multiple conditions, such as functional impairment (FI), cognitive impairment or neurocognitive disorder (NCD), pressure ulcer, falls, and frailty—health conditions which cannot be grouped into any diseases [3]. These conditions are manifested as the results of accumulated insults from both internal and external factors over time. Empirical studies have found that 21.5% (range: 10.4%–53.9%), [8] 19.0% (range: 5.1%–41.0%), [9] and 8.6%–14.5% [10] of older adults experience FI, NCD, and frailty, respectively. According to the World Health Organization (WHO) report, in low-income countries, the majority of elderly people living with frailty are concentrated among those with lower socioeconomic status (SES) [10].

Socioeconomic inequality is a critical determinant of health [11]. Apart from accumulating health impairment with aging, adults are also exposed to socioeconomic vulnerability and become more inequitable. Recent empirical studies have found that income inequality tends to be greater when people age; [1, 12] ones who live in poverty tend to be poorer when they age, while richer people tend to gain continuous profits. Further, poor socioeconomic status is evidently related to poorer health outcomes in all age groups [13, 14] and accelerates aging impairments, including physical capacity; sensory, physiological, and cognitive functions; emotions; and social domains [15]. Further, several previous studies have found association between socioeconomic status and frailty, [10] cognitive impairment, [16, 17] and impaired functional capacity [18, 19]. Therefore, older adults are vulnerable to inequitable health issues.

Health inequality is a major global health challenge and refers to the conceptual inequality of health between individuals [20]. Reducing inequality within and among countries is one of the United Nations Sustainable Development Goals (SDG 10) by 2030; [2] thus, learning about health inequality at different socioeconomic levels can help policymakers implement health interventions to reduce the gap in health between rich and poor people [20]. There are several ways to measure health inequalities. Simple comparison methods, such as using means or proportions between two groups, can provide simple measures; however, they exclude comparisons among some subgroups. Another way to assess health inequality is to compare the health levels of all individuals or subgroups using more complex techniques, such as the concentration index, whose values provide fine-grained information of inequality at the individual level —the extent a health outcome is distributed among different individual SES levels [21]. One study in Thailand showed concentration of aging among the poor, [22] while another study reported concentration of favorable self-rated health among the better-off and explained that the inequitable distribution of health was mainly explained by certain avoidable factors, such as income, residence, region, medical insurance, employment [23]. Another study in China applied the concentration index to illustrate changes in the inequality of FI in the elderly and found that the inequality increased from 2008 to 2018, and such a change was explained by out-of-pocket payments (OOP) for medical bills, living in rural regions, and the density of skilled health workers [24].

Equitable implementation of policies and interventions in GS requires information on the SES of the target population, but such information is inconsistent in Thailand. Earlier studies have shown the degree of socioeconomic inequality regarding aging and the general health of older individuals in the last decade; [22] however, there is still a lack of information about inequality in GS distribution. Further, priority setting for capacity building and policy development requires more in-depth information on the extent each socioeconomic determinant contributes to inequality. In Thailand, information of various health determinants and geriatric health problems have been reported in a series of national health examination surveys (NHES). These surveys were used to illustrate the distribution of health conditions across the country and help public health authorities devise policies to improve health equity. The NHES collects data on GS, including cognitive and functional impairment, socioeconomic status, and some GS phenotypes, which are separately analyzed and reported. Therefore, we aimed to further examine potential links between socioeconomic inequalities and GS and apply a decomposition method to illustrate the socioeconomic and sociodemographic determinants that contributed to inequalities associated with GS.

## Methods

### Study setting and participants

We performed a secondary data analysis of the Fifth National Health Examination Survey (NHES-5), conducted in 2013. The NHES has been conducted every 5 years since 1991. The NHES-5 is a nationwide cross-sectional survey that uses a multistage, stratified sampling of the Thai population. The survey included 32,400 participants of all ages, starting from one year of age, in Bangkok and in 20 provinces of four regions in Thailand: namely, central, northern, northeastern, and southern regions. Well-trained fieldworkers conducted face-to-face structured interviews and physical examinations. A total of 7,365 participants aged 60 years old and older were included in the NHES-5 survey. Those (1,904 participants in total) who did not report household assets (i.e., missing wealth index values) were excluded from the analysis.

### Variables

**Sociodemographic variables.** Previous studies have reported various potential characteristics associated with socioeconomic inequality in the presentation of GS [24–26]. We used age, gender, marital status, education, working status, health insurance, and region of residence, which were investigated in the NHES-5 [27]. Health insurance consists of three major health schemes: Universal Health Coverage scheme (UCS), Social Security Scheme (SSS), Civil Servant Medical Benefit Scheme (CSMBS), and other minor health schemes. Age and gender were considered as "need variables," representing the sources of variation in healthcare that are not considered to be "unfair" while other socio-demographic variables were treated as "non-need variables" in the analysis of horizontal inequality [28].

### Geriatric syndrome

We included three conditions of geriatric syndrome as available in the data NHES-5 in this study:

*Cognitive impairment*: The Thai version of the Mini-Mental State Examination 2002 [29] was applied. It covers orientation, registration, attention, memory, speech, understanding, abstract thinking, and visuoconstruction. Three cut-off values for impairment were applied based on the participants' highest degree of education (maximum score 30): ≤14, ≤17, and

$\leq$22 points for no formal education (or inability to read and write), primary school, and higher than primary school, respectively.

*Functional impairment*: Functional capacity was assessed using Barthel's Activities of Daily Living index [30]. We applied the NHES-5 criteria to define dependency. Elderly individuals who met at least one of the following two indicators were considered to have functional dependency: (1) requiring partial or total support in two or more daily activities, and (2) having urinary or fecal incontinence.

*Frailty*: We applied frailty criteria from the WHO study in low-to-middle-income countries [10]. The criteria assessed respondents based on 5 components: (1) body mass index (BMI) being the lowest quintile in the country, (2) low-grip strength (age and sex specific cut points), (3) physical activity as assessed by Global Physical Activity Questionnaire (GPAQ) < 600 MET/min, (4) 4-meters timed walk test < 0.5 meter per second, and (5) exhaustion (using structured interview: "For the past 12 months, have you often felt fatigue, or have easily been tired or bored?"). Participants were categorized based on the frailty score: robust (score 0), pre-frail (score 1–2), and frail (score $\geq$ 3).

## Socioeconomic status (SES)

The wealth index is used to represent individual SES variables. The index was deemed more appropriate for SES as it can reflect a more stable economic status and the collective wealth of the entire family. The wealth index was generated based on the method suggested in MEASURE DHS+ surveys [31]. The survey participants were asked if they owned any of the following: bed, air conditioner, electric water boiler, washing machine, microwave, personal computer, house telephone, car, and flushing toilet. The asset variables were converted to indicator variables, standardized by computing z-scores, and used to calculate factor loadings from the principal component analysis. The indicator values were then multiplied by the loadings and summed for all assets to produce the household index value.

## Inequality measurements

**Concentration index (C).**   The concentration index is one of the most complex methods for analyzing health inequalities [21]. According to Kakwani (1997) and O' Donnell et al. (2007), [28, 32] concentration index can be computed using three parameters: health outcome mean ($\mu$), health outcome of an individual ($y_i$), in connection with its position ($i$) on the full population SES, and fractional rank ($r_i$): the ratio between individual SEP and the population size ($i$/N) based on the full socioeconomic distribution of the population. The concentration index was calculated using the following formula:

$$C = \frac{2}{\mu} Cov(y_i r_i) \tag{1}$$

Eq 1 shows that the index is computed using the covariance between individual health outcomes and their fractional rank. The index typically ranges from -1 to 1, with a positive value indicating a concentration of the outcome among groups with higher SES (the outcome accumulating more among well-off individuals); a negative value suggesting that individuals in the lower SES group are more likely to experience the outcome than the affluent; and a value of zero indicating no socioeconomic-related inequality. However, owing to limitations in assessing the weights of the outcome across the entire SES distribution, it is recommended to use C in conjunction with the concentration curve.

When dealing with categorical health outcome variables, it is advisable to dichotomize the health measurement for the calculation. Further, a normalized C can be computed by dividing

the original C by 1 minus the mean if one intends to compare the inequality between two or more settings [28]. For the NHES-5, the design necessitates the use of weighted fractional rank [28]. According to Eq 2, sampling weights are incorporated as represented by a fraction value of an individual weight to total weights scaled sum to one ($w$): $w_j$ (value of an individual whose rank is 1 rank lower than $i$) and $w_i$ (value of an individual at position $i$).

$$r_i = \sum_{J=0}^{i-1} w_j + \frac{w_i}{2} \tag{2}$$

**Decomposition of concentration index.** Decomposition of concentration index was used to illustrate the contribution of different socioeconomic and sociodemographic variables to health outcome inequality ($C$). Eq 3 shows a linear addition model of the contributions of several variables: the need ($x$) and non-need ($z$) variables. This representation is based on probit regression model and marginal effects were evaluated at each variable's mean of $j^{th}$ need variable and $k^{th}$ non-need variable ($\beta_j^m$ and $\gamma_k^m$) [28]. Each contribution (or absolute contribution: AC) is a product of the variable's concentration index and its elasticity (the product of marginal effects, variable's mean, and health outcome's mean: $\frac{\beta_j^m \bar{x}_j}{\mu}$ and $\frac{\gamma_k^m \bar{z}_k}{\mu}$). The last term is expressed as the product of the generalized concentration index of the residuals ($GC_\varepsilon$) and the health outcome mean. The closer the last term is to zero, the better its ability to explain health inequality using the decomposition equation. Further, the percentage contribution (%CC), which is the percentage value of each AC relative to the $C$, can be used to further illustrate the size and direction of each variable's contribution to inequality. Positive %CC values indicate that the variables contributed to a greater degree of health outcome inequality. Further, this decomposition method allows us to evaluate the horizontal equity using indirectly standardized concentration index of health outcome ($C^{is}$) which is acquired by subtracting $C$ by the need contributions [28].

$$C = \sum C_j \left( \frac{\beta_j^m \bar{x}_j}{\mu} \right) + \sum C_k \left( \frac{\gamma_k^m \bar{z}_k}{\mu} \right) + \frac{GC_\varepsilon}{\mu} \tag{3}$$

## Statistical analysis

Sampling weights were applied to all the analyses. Prevalence was reported as a weighted percentage. Unstandardized and indirectly standardized concentration indices were used to describe socioeconomic inequality in all presentations of GS. All variables were transformed into indicator variables. Differences in proportions between the groups were tested using the chi-square test with Wald statistics. To decompose inequality, a probit model was applied in which marginal effects evaluated at the mean of each variable were used as input coefficients for the decomposition analyses. Bootstrapping with 10,000 replications was additionally used to acquire additional 95% confidence interval of the concentration index as its distributional information is limited [33]. R software version 4.2.2 with packages survey, IC2, stringr, epicalc, marginaleffects, and boot was used.

## Ethical considerations

The participants had been given detailed information about the study procedures and anonymity assurance before they signed the informed consent document. We analyzed de-identified secondary data from the NHES-5 after receiving ethical approval from the Human

Research Ethics Committee, Faculty of Medicine, Prince of Songkla University (project number: REC 62-054-18-1).

## Results

We included 5,461 participants in the analysis. Table 1 shows that the majority of the participants were 60–79 years old, were living with their spouses, and had a primary school or lower education. Approximately 40% of participants worked actively, while the remaining majority were either unemployed or housemakers. More than 4,000 individuals had UCS (85.3%), whereas 220, 414, and 610 individuals (3.3%, 5.7%, and 10.0%) had SSS, CSMBS, and other health insurance plans (e.g., private insurances and state enterprise schemes), respectively. Six hundred and forty-five individuals reported that they had used two or more health insurance plans. The southern region was the least common area of residence of the study participants (6.7%). The participants' SES were relatively balanced among the three levels.

**Table 1. Demographic characteristics of the participants.**

| Variable | n | Weighted % (95% CI) |
|---|---|---|
| Age group | | |
| 60–69 years | 3,264 | 56.9 (53.8, 59.9) |
| 70–79 years | 1,669 | 30.0 (28.7, 31.3) |
| $\geq 80$ years | 528 | 13.1 (10.5, 15.7) |
| Gender | | |
| Male | 2,392 | 45.4 (44.7, 46.2) |
| Female | 3,069 | 54.6 (53.8, 55.3) |
| Marital status | | |
| Married | 3,299 | 60.8 (57.9, 63.8) |
| Divorced or separated | 385 | 6.4 (5.4, 7.4) |
| Widowed | 1,456 | 27.0 (24.5, 29.5) |
| Single | 321 | 5.7 (4.2, 7.2) |
| Education | | |
| $\geq$ Secondary school | 868 | 12.3 (5.5, 19.0) |
| $\leq$ Primary school | 4,593 | 87.7 (81.0, 94.5) |
| Working status | | |
| Working/ employed | 2,171 | 40.4 (36.8, 44.0) |
| Unpaid work/ unemployed | 3,290 | 59.6 (56.0, 63.2) |
| Health insurance | | |
| UCS | 4,481 | 85.3 (81.2,89.4) |
| SSS | 220 | 3.3 (1.5, 5.2) |
| CSMBS | 414 | 5.7 (3.5, 8.0) |
| Other | 610 | 10.4 (9.0, 11.9) |
| SES | | |
| Tercile1 (poorest) | 1,510 | 33.5 (25.6, 41.5) |
| Tercile2 (middle) | 1,718 | 33.3 (28.2, 38.4) |
| Tercile3 (richest) | 2,233 | 33.1 (22.7, 43.5) |
| Region of residence | | |
| Bangkok | 732 | 10.7 (0.0, 30.4) |
| North | 1,565 | 24.7 (15.5, 33.9) |
| Central and West | 1,266 | 23.1 (9.5, 36.7) |
| Northeast | 1,360 | 34.9 (21.1, 48.7) |
| South | 538 | 6.7 (1.4, 12.0) |

FI was the most common phenotype of GS (20.0% vs. 8.6% and 6.9% for NCD and frailty, respectively). Being in a higher age group, being female, being widowed, having a primary or lower education, and being unemployed were common characteristics of participants who had experienced GS. Age, marital status, and work status were significant factors associated with GS, with a higher prevalence among older age groups, widowed individuals, and the unemployed. In addition to age, a lower SES appears to be associated with a higher prevalence of some GS conditions. In the lowest SES group, the prevalence of NCD, FI, and frailty was 12.6%, 21.9%, and 8.3%, respectively (Table 2).

Our decomposition analysis produced AC and %CC which represent the extent of inequality in each variable that was responsible for each GS phenotype. Table 3 shows unstandardized and indirectly standardized concentration index of GS and decomposes inequalities into subcomponent contributions from both need- and non-need variables. The magnitude of inequalities in FI ($C^{is}$ = -0.068) and frailty ($C^{is}$ = -0.092) were relatively smaller than that in NCD ($C^{is}$ = -0.182). The concentration curves illustrating the distributions of the three GS phenotypes are provided in S1 Fig. Most cases of inequality in GS: 64.7%, 78.6%, and 53.1% of NCD, FI, and frailty, respectively, were explained by non-need factors. Regarding the need variables that partially determined GS inequalities, being older played a relatively greater role in the inequalities of geriatric syndrome than gender (11.8% vs 0.3%, 13.5% vs 1.9%, and 23.5% vs -0.1%, in NCD, FI, and frailty, respectively). Education made the highest contribution to the unequal distribution of NCD, and the higher education of the population contributed to lower inequality (-12.2%). Small contributions of the working status (working vs being unemployed) were found among the GS, but such contributions were uniform in terms of the direction (-2.0%, -1.7%, and -5.1% in NCD, FI, and frailty, respectively). Thus, promoting work opportunity might lead to a more equitable GS distribution in the population. Apart from those factors, the regions of participants' residences was another non-need variable which made 30.0% and 23.7% contribution to widening inequality of FI and lowering inequality of frailty, respectively. The concentration index of each need- and non-need variable and a decomposition analysis of the normalized concentration index are provided in S1 File Supplementary tables.

## Discussion

We analyzed the data of individuals aged 60 years and above from the NHES-5 to investigate socioeconomic inequalities of GS using concentration indices and further explained inequalities by using the decomposition method to show the contributions of socioeconomic and sociodemographic determinants to the unequal distributions of GS between the poor and better-off elderly [28]. We found that GS was distributed in a dose-response pattern with age and was more common in individuals who were females and widows, had lower education, and were unemployed. In regard to SES, a higher prevalence of GS was found among individuals with lower SES status, falling within the first and second terciles of SES. After standardizing inequalities by need variables, we found that all GS phenotypes are significantly concentrated among poor individuals and were mainly explained by non-need variables. The decomposition results could be useful for policymakers who want to see the effect of factors whose distribution could be considered unfair on the inequitable distributions of GS phenotypes and prioritize these factors when devising health policies for more equitable health of the population.

Impaired cognitive function, lack of functional capacity, and frailty can be burdensome for affected individuals and their families (as the elderly become dependent) and cause a great financial burden on the healthcare system. Several studies have shown that these GS phenotypes were associated not only with the increased likelihood of poorer health outcomes and mortality in the elderly, [34–36] but also with an increased caregiver burden, [37] distress

**Table 2. Distribution (weighted %) of cognitive impairment, functional impairment, and frailty in socioeconomic and socio-demographic characteristics.**

| Variable | Cognitive impairment | Functional impairment | Frailty |
|---|---|---|---|
| Overall | 8.6 (7.3, 10.0) | 20.0 (18.3, 21.7) | 6.9 (5.5, 8.2) |
| Age group | *p* < 0.001 | *p* = 0.008 | *p* < 0.001 |
| 60–69 years | 4.8 (3.8, 5.8) | 15.5 (12.7, 18.3) | 3.0 (2.2, 3.8) |
| 70–79 years | 9.1 (7.5, 10.8) | 23.8 (21.8, 25.9) | 9.7 (7.4, 12) |
| ≥ 80 years | 24.2 (20.2, 28.2) | 30.5 (25.4, 35.6) | 17.4 (13.2, 21.6) |
| Gender | *p* = 0.003 | *p* < 0.001 | *p* = 0.250 |
| Male | 6.8 (5.7, 8.0) | 15.7 (13.6, 17.9) | 6.2 (4.4, 8.0) |
| Female | 10.1 (8.5, 11.8) | 23.5 (21.4, 25.6) | 7.5 (5.7, 9.2) |
| Marital status | *p* < 0.001 | *p* = 0.026 | *p* = 0.070 |
| Married | 6.4 (5.1, 7.7) | 17.9 (15.3, 20.5) | 5.6 (4.3, 6.9) |
| Separated | 4.6 (1.1, 8.2) | 20.0 (15.1, 24.8) | 6.7 (3.8, 9.5) |
| Widowed | 13.2 (10.8, 15.6) | 25.7 (23.1, 28.2) | 9.9 (7.0, 12.9) |
| Single | 15.3 (8.8, 21.9) | 14.7 (7.4, 22.0) | 6.3 (0.2, 12.5) |
| Education | *p* = 0.805 | *p* = 0.024 | *p* = 0.011 |
| ≥ Secondary school | 8.4 (6.6, 10.3) | 13.7 (11.1, 16.4) | 4.4 (2.7, 6.1) |
| ≤ Primary school | 8.7 (7.3, 10.1) | 20.8 (19, 22.7) | 7.2 (5.8, 8.6) |
| Working status | *p* < 0.001 | *p* < 0.001 | *p* <0.001 |
| Working | 3.8 (3.0, 4.6) | 15.4 (13.0, 17.7) | 2.3 (1.2, 3.5) |
| Unemployed | 11.9 (9.8, 14.1) | 23.1 (21.2, 25) | 10.0 (7.6, 12.3) |
| Having UCS | *p* = 0.017 | *p* = 0.927 | *p* = .248 |
| Yes | 9.2 (7.7, 10.6) | 20.0 (18.1, 21.8) | 7.1 (5.7, 8.6) |
| No | 5.5 (3.7, 7.3) | 20.1 (11.6, 22.6) | 5.5 (3.1, 7.9) |
| Having SSS | *p* = 0.057 | *p* = 0.765 | *p* = 0.273 |
| Yes | 2.7 (0.3, 5.0) | 21.2 (14.2, 28.2) | 3.7 (-0.3, 7.7) |
| No | 8.2 (7.5, 10.2) | 19.9 (18.1, 21.7) | 6.9 (5.6, 8.4) |
| Having CSMBS | *p* = 0.380 | *p* = 0.211 | *p* = 0.913 |
| Yes | 7.1 (3.8, 10.4) | 15.9 (10.2, 21.6) | 6.7 (3.0, 10.4) |
| No | 8.7 (7.3, 10.1) | 20.2 (18.5, 21.8) | 6.9 (5.5, 8.3) |
| Having other insurance plans | *p* = 0.012 | *p* = 0.050 | *p* = 0.245 |
| Yes | 3.5 (1.5, 5.4) | 13.5 (8.5, 18.6) | 5.1 (1.8, 8.4) |
| No | 9.2 (7.7, 10.8) | 20.7 (18.8, 22.6) | 7.1 (5.8, 8.4) |
| SES | *p* = 0.030 | *p* = 0.005 | *p* = 0.075 |
| Tercile1 (poorest) | 12.6 (9.6, 15.7) | 21.9 (19.0, 24.7) | 8.3 (6.5, 10.1) |
| Tercile2 (middle) | 7.9 (6.1, 9.7) | 22.2 (19.0, 25.5) | 7.0 (5.4, 8.7) |
| Tercile3 (richest) | 5.4 (4.5, 6.3) | 15.8 (14.3, 17.3) | 5.3 (3.6, 6.9) |
| Region | *p* = 0.016 | *p* = 0.002 | *p* = 0.051 |
| Bangkok | 7.2 (7.2, 7.3) | 16.1 (16, 16.2) | 6.7 (6.6, 6.7) |
| North | 10.8 (9.7, 11.9) | 20.9 (17.7, 24.1) | 8.6 (6.4, 10.8) |
| Central and West | 6.1 (5.0, 7.3) | 13.3 (10.3, 16.4) | 7.7 (2.7, 12.6) |
| Northeast | 8.7 (5.4, 12.1) | 23.4 (19.1, 27.6) | 5.3 (5.0, 5.6) |
| South | 11.0 (10.3, 11.7) | 28.0 (24.6, 31.5) | 6.5 (3.7, 9.3) |

among families, [37] and increased personal and healthcare system expenses, [38, 39] bringing significant burden to the society and families. Therefore, several studies have investigated the possible determinants of these phenotypes of GS. According to previous studies, multiple socioeconomic and sociodemographic factors, including being females, having lower education, [19, 40, 41] having a lower income, [19, 41] and unemployment, [19] contributed to the

**Table 3. Concentration index and contribution of socioeconomic and sociodemographic determinants to cognitive impairment, functional dependency, and frailty.**

| | Cognitive impairment | | Functional impairment | | Frailty | |
|---|---|---|---|---|---|---|
| C (95% CI) | -0.220 (-0.316, -0.140) | | -0.082 (-0.131, -0.037) | | -0.130 (-0.233, -0.045) | |
| $C^{is}$ (95% CI) | -0.182 (-0.208, -0.076) | | -0.068 (-0.096, -0.021) | | -0.092 (-0.096, -0.023) | |
| | **AC** | **%CC** | **AC** | **%CC** | **AC** | **%CC** |
| **Need variables** | | | | | | |
| Age | -0.023 | 11.8 | -0.010 | 13.5 | -0.026 | 23.5 |
| Gender | -0.001 | 0.3 | -0.001 | 1.9 | <0.001 | -0.1 |
| **Subtotal** | **-0.023** | **11.8** | **-0.010** | **13.5** | **-0.026** | **23.5** |
| **Non-need variables** | | | | | | |
| Marital status | -0.005 | 2.4 | <0.001 | 0.8 | <0.001 | -0.2 |
| Education | 0.024 | -12.2 | -0.007 | 9.7 | -0.011 | 10.1 |
| Working status | 0.004 | -2.0 | 0.001 | -1.7 | 0.006 | -5.1 |
| UCS | -0.015 | 7.8 | 0.009 | -11.3 | -0.006 | 5.1 |
| SSS | -0.003 | 1.7 | 0.004 | -5.1 | -0.003 | 2.3 |
| CSMBS | 0.006 | -3.3 | -0.003 | 4.2 | 0.004 | -3.6 |
| Other insurance | -0.006 | 3.2 | -0.004 | 5.4 | <0.001 | 0.2 |
| SES | -0.123 | 63.0 | -0.035 | 46.6 | -0.076 | 68.0 |
| Region | -0.008 | 4.2 | -0.024 | 30.0 | 0.027 | -23.7 |
| **Subtotal** | **-0.126** | **64.7** | **-0.059** | **78.6** | **-0.059** | **53.1** |
| Unexplained C | 0.094 | | 0.197 | | -0.009 | |

Remarks: C, concentration index; $C^{is}$ indirectly standardized concentration index; AC, Absolute contribution to the concentration index; %CC, Percentage contribution to the concentration index.

increased likelihood of NCD, FI, and frailty in older individuals. Our findings contribute to the literature by showing that the socioeconomic inequalities in NCD, FI, and frailty among the Thai population were also explained by age, education, and working status. While age was considered an inherent biological determinant of GS, education and employment can be seen as factors that were potentially unfairly distributed.

The current study showed that active work contributed to a more equitable distribution of GS in the older population. The multifactorial model explained GS by interactive accumulated biological risks across an individual's lifetime and linkage with psychosocial factors [5]. Further, some studies found that employment could reduce the risk of cognitive impairment in later life [42, 43]. We speculate that the narrowing gap in GS between well-off and poor individuals could be caused by the presence of working activities. However, a previous study found that only time away from work caused by early unemployment, sickness, and voluntarily homemaking increased the risk of NCD in older individuals while work leave for training and maternity might reduce the risk of NCD [42]. Thus, caveats should be considered that promoting active-working status at some points before the retirement age might reduce equity gaps only in some cases of unemployed individuals. Nowadays, the increase in the older age proportion in Thailand has created challenges to the labor market despite narrowing gaps in education opportunities in younger generations [44]. An aging society shifts the workforce by shrinking proportion of working-age individuals and expanding the ratio of older candidates [45]. Further, older individuals' working or job performance can be limited by various age-related changes, including cognitive impairment [46–48]. Therefore, from the market perspective, additional care responsibilities for the elderly could be required from employers. Owing to these concerns, a policy that helps older people access work opportunities or extend their

working life should be considered to reduce the inequitable distribution of GS. Nowadays, the impact of age-related complexities on work capacity can be handled using several modern technologies, such as corrective eyewear, hearing aids, and other assistive technologies [49].

The national health insurance system is also key to achieving universal health coverage and equitable health for the population [50]. We found that socioeconomic inequality in GS was partially attributed to different Thai health schemes. UCS was the insurance scheme, which was a co-paid (0.1$/ visit) medical insurance originally provided for unemployed, older, and poor people [51]. Conversely, CSMBS was relatively superior to the other health schemes. Prevention and health promotion programs were only available in UCS [51]. Although UCS had covered more than 80% of eligible population in 2013, the majority (60%) of the UCS population required OOP [52]. In our study, more than 80% the participants had UCS, the insurance scheme which contributed 7.8% increase, 11.3% reduction, and 5.1% increase in the socioeconomic inequality of NCD, FI, and frailty, respectively. Similarly, the contributions from SSS and CSMBS were mixed among the three phenotypes. Conversely, minor health schemes—for example, insurance for government and state officers—were part of non-need factors which contributed to more inequitable distribution of all GS phenotypes. A previous study conducted among the Chinese elderly [23] emphasized that disparities and fragmentations of the health insurance scheme significantly impacted the overall self-rated health of the elderly whereas another study in rural China [53] found no significant correlation between health and health insurance. We argue that the strategy to deal with these bidirectional contributions and overall elderly health regarding GS, which enables individuals' access to a certain health scheme (resulting in either more or less equitable distributions in certain GS phenotypes) might not be as effective as enhancing equality of health scheme access between the rich and the poor (making inequalities converge towards zero). Apart from such concerns, it is worth noting that interventions such as Multicomponent Targeted Intervention (MTI) and frailty intervention (FTI) were designed to prevent GS and were found to be efficacious and cost-effective [54, 55]. Thus, we suggest that some of these preventive interventions be considered in the universal health coverage schemes, as they might help reduce gaps between rich and poor elderly with GS.

To the best of our knowledge, this is the first study to use data from a population-based survey to measure socioeconomic inequalities of the three GS phenotypes and to approximate the probable values of inequalities using bootstrap confidence intervals. We also explained that some sociodemographic and socioeconomic factors potentially contributed to inequity of individuals with GS. However, this study had several limitations. First, the decomposition analyses of the concentration indices only provided the magnitude and direction of the contribution to the existing inequity during the study period owing to the cross-sectional design. Thus, the contributions presented do not imply that a single factor causes a certain percentage of inequality in the GS. Second, we analyzed secondary data GS phenotypes presented in the Thai population, which were available from the NHES-5. Thus, the current findings illustrate situations of inequality only in NCD, FI, and frailty, and might not be generalized to all GS phenotypes, such as decubitus ulcers and polypharmacy. Therefore, we suggest that the interpretation and generalization of the study results should be performed carefully.

## Conclusion

Cognitive impairment, functional decline, and frailty were significantly more prevalent in the poor elderly group, of which the observed socioeconomic inequality was relatively larger among those with cognitive impairment. Employment status and health insurance schemes can be treated as potentially modifiable non-need factors that required attention from health policymakers. Distributions of employment status provided uniform explanation for the

widening inequality of all GS phenotypes. Therefore, the study emphasizes the importance of working opportunity in modifying inequitable GS in old age. Bidirectional contributions from inequitable health insurance schemes suggest that equalizing access or consolidating health insurance schemes could be a more promising strategy for the country's universal health coverage than providing access to any one of the schemes. Inequitable cognitive impairment, employment, and access to insurance schemes required further attention from policymakers to establish a healthy aging society. Nevertheless, future study should evaluate the impact of these socioeconomic contributions overtime and investigate the effectiveness of public health interventions which have been implemented.

## Supporting information

**S1 Fig. Concentration curves of the 3 geriatric syndrome phenotypes.**
(TIF)

**S1 File. Supplementary tables.**
(PDF)

## Acknowledgments

NHES-5 was conducted by the National Health Examination Survey Office, Health Systems Research Institute, Thailand. We would like to thank Associate Professor Polathep Vichitkunakorn, Mr. Kittisak Chumalee and Ms. Yongmei Jin for their constructive statistical and programming suggestions.

## Author Contributions

**Conceptualization:** Supakorn Sripaew, Sawitri Assanangkornchai.

**Data curation:** Supakorn Sripaew, Sawitri Assanangkornchai, Jiraluck Nontarak.

**Formal analysis:** Supakorn Sripaew, Sawitri Assanangkornchai, Jiraluck Nontarak.

**Investigation:** Sawitri Assanangkornchai, Jiraluck Nontarak, Suwat Chariyalertsak, Pattapong Kessomboon, Surasak Taneepanichskul, Nareemarn Neelapaichit, Wichai Aekplakorn.

**Methodology:** Supakorn Sripaew, Sawitri Assanangkornchai, Jiraluck Nontarak, Suwat Chariyalertsak, Pattapong Kessomboon, Surasak Taneepanichskul, Nareemarn Neelapaichit, Wichai Aekplakorn.

**Project administration:** Sawitri Assanangkornchai, Jiraluck Nontarak, Suwat Chariyalertsak, Pattapong Kessomboon, Surasak Taneepanichskul, Nareemarn Neelapaichit, Wichai Aekplakorn.

**Resources:** Sawitri Assanangkornchai, Suwat Chariyalertsak, Pattapong Kessomboon, Surasak Taneepanichskul, Nareemarn Neelapaichit, Wichai Aekplakorn.

**Supervision:** Sawitri Assanangkornchai, Jiraluck Nontarak, Wichai Aekplakorn.

**Validation:** Supakorn Sripaew, Sawitri Assanangkornchai, Jiraluck Nontarak, Suwat Chariyalertsak, Pattapong Kessomboon, Surasak Taneepanichskul, Nareemarn Neelapaichit, Wichai Aekplakorn.

**Visualization:** Supakorn Sripaew, Sawitri Assanangkornchai, Jiraluck Nontarak.

**Writing – original draft:** Supakorn Sripaew, Sawitri Assanangkornchai.

**Writing – review & editing:** Supakorn Sripaew, Sawitri Assanangkornchai, Jiraluck Nontarak, Suwat Chariyalertsak, Pattapong Kessomboon, Surasak Taneepanichskul, Nareemarn Nee-lapaichit, Wichai Aekplakorn.

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
