## [Decision Letter · Decision Letter 0]

19 Sep 2023

PONE-D-23-15911Socioeconomic inequalities associated with Geriatric syndrome in Thailand: the results of Fifth National Health Examination SurveyPLOS ONE

Dear Dr. Sripaew,

Thank you for submitting your manuscript to PLOS ONE. After careful consideration, we feel that it has merit but does not fully meet PLOS ONE’s publication criteria as it currently stands. Therefore, we invite you to submit a revised version of the manuscript that addresses the points raised during the review process.

We look forward to receiving your revised manuscript.

Kind regards,

Peivand Bastani

Academic Editor

PLOS ONE

We will update your Data Availability statement on your behalf to reflect the information you provid

Reviewers' comments:

Reviewer's Responses to Questions

**Comments to the Author**

1. Is the manuscript technically sound, and do the data support the conclusions?

Reviewer #1: Yes

Reviewer #2: Yes

2. Has the statistical analysis been performed appropriately and rigorously? 

Reviewer #1: Yes

Reviewer #2: Yes

3. Have the authors made all data underlying the findings in their manuscript fully available?

Reviewer #1: Yes

Reviewer #2: Yes

4. Is the manuscript presented in an intelligible fashion and written in standard English?

Reviewer #1: Yes

Reviewer #2: Yes

5. Review Comments to the Author

Reviewer #1: Many thanks for this very interesting manuscript working on an important health care topic.

I have only some minor comments, which can be found below.

Abstract:

- The authors propose some applied suggestions for reducing inequality.

- Please provide numerical values for concentration index and decomposition.

Background:

- It is need to be explained what are the gaps in the literature that authors wanted to address before reporting this research- in other words, what is already known and what does this article add to the literature?

Results:

Table 1 must be reported in the separate page.

Discussion:

Discussion should focus on critical interpretation of the findings. The study finding must be compared with existing agreed and disagreed cases. In this study, I don't see any comparison with disagreed cases.

The conclusion section is pretty short. Maybe that could be extended

Reviewer #2: Thanks for giving me the opportunity to review this interesting manuscript. I have some comments about this manuscripts

1. The title and the introduction of the manuscript sound the social science but the methodology showed the high level of statistical idea. It will be better to explain the idea of index and the statistical methods that show clearer idea of GS and inequlity.

2. Table 2 presented the symbol "*" to illustrate the level of p-value significant. I recommend to present with p-value.

3. For Table 3 and the text explanation, this section was important for answer the research question. However, it hard to understand for less knowledge of statistics. It will be better to explain more for application of this results.

4. For the limitation, from "Thus, the current findings may not represent all GS phenotypes..." What types of phenotypes that we can generalized.

Thanks.

6. PLOS authors have the option to publish the peer review history of their article (what does this mean?). If published, this will include your full peer review and any attached files.

Reviewer #1: No

Reviewer #2: No

---

## [Author Response · Author response to Decision Letter 0]

1 Mar 2024

Reviewer 1

1.Abstract:

- The authors propose some applied suggestions for reducing inequality.

- Please provide numerical values for concentration index and decomposition. 

Response: We added numerical values of indirectly standardized concentration index and percentage of contribution to the concentration index in the relevant sections. 

Page 2 line 38

“Contributions to the concentration index (CC)—contribution to a more or less equitable GS distribution—were decomposed and shown in terms of percentage and direction All GS phenotypes were found to be concentrated in the elderly poor (Cis of FI, frailty, and NCD = - 0.068, -0.092, and -0.182, respectively). Work status contributes to a more equitable GS distribution in all the phenotypes (%CC in FI, frailty, and NCD = - 1.7%, -5.1%, and -2.0%, respectively), whereas types of insurance schemes made bidirectional contributions to the equity of GS.”

2.Background:

- It is need to be explained what are the gaps in the literature that authors wanted to address before reporting this research- in other words, what is already known and what does this article add to the literature? 

Response: We emphasize the gap of knowledge in the latter part of our introduction by statements as following. 

Page 4 line 96

“Equitable implementation of policies and interventions in GS requires information on the SES of the target population, but such information is inconsistent in Thailand. Earlier studies have shown the degree of socioeconomic inequality regarding aging and the general health of older individuals for past decades; however, there is still a lack of information about inequality in GS distribution. Further, priority setting for capacity building and policy development requires more in-depth information on the extent each socioeconomic determinant contributes to inequality.”

3.Results:

Table 1 must be reported in the separate page. 

Response: We modified the descriptions pertaining Table 1 information to elaborate details relevant to the discussions and reported the Table in a separate page. The lower bound of the region estimation (Bangkok) was also changed from -9.1% to 0.0% as a legitimate value for the parameter’s bound (percentage). 

Page 10 line 233

“Approximately 40% of participants worked actively, while the remaining majority were either unemployed or housemakers. More than 4,000 individuals had UCS (85.3%), whereas 220, 414, and 610 individuals (3.3%, 5.7%, and 10.0%) had SSS, CSMBS, and other health insurance plans (e.g., private insurances and state enterprise schemes), respectively. Six hundred and forty-five individuals reported that they had used two or more health insurance plans. The southern region was the least common area of residence of the study participants (6.7%). The participants’ SES were relatively balanced among the three levels.”

4.Discussion:

Discussion should focus on critical interpretation of the findings. The study finding must be compared with existing agreed and disagreed cases. In this study, I don't see any comparison with disagreed cases. 

Response: We added discussion points to compare the current findings with some disagreed cases regarding the impact of health schemes to elderly health and discuss them with the current findings. 

Page 17 line 330

“The current study showed that active work contributed to a more equitable distribution of GS in the older population. The multifactorial model explained GS by interactive accumulated biological risks across an individual’s lifetime and linkage with psychosocial factors.5 Further, some studies found that employment could reduce the risk of cognitive impairment in later life.46,47 We speculate that the narrowing gap in GS between well-off and poor individuals could be caused by the presence of working activities. However, a previous study found that only time away from work caused by early unemployment, sickness, and voluntarily homemaking increased the risk of NCD in older individuals while work leave for training and maternity might reduce the risk of NCD.47 Thus, caveats should be considered that promoting active-working status at some points before the retirement age might reduce equity gaps only in some cases of unemployed individuals.”

And in another paragraph,

Page 18 line 365

“A previous study conducted among the Chinese elderly24 emphasized that disparities and fragmentations of the health insurance scheme significantly impacted the overall self-rated health of the elderly whereas another study in rural China57 found no significant correlation between health and health insurance. We argue that the strategy to deal with these bidirectional contributions and overall elderly health regarding GS, which enables individuals’ access to a certain health scheme (resulting in either more or less equitable distributions in certain GS phenotypes) might not be as effective as enhancing equality of health scheme access between the rich and the poor (making inequalities converge towards zero).”

Some statements were also modified to clarify the direction of association between risk factors and geriatric syndrome as follows:

Page 16 line 319

“Therefore, several studies have investigated the possible determinants of these phenotypes of GS. According to previous studies, multiple socioeconomic and sociodemographic factors, including being females, having lower education,20,44,45 having a lower income,20,45 and unemployment,20 contributed to the increased likelihood of NCD, FI, and frailty in older individuals.”

5.The conclusion section is pretty short. Maybe that could be extended 

Response: We extended our conclusion to give a more detailed summary as well as suggestions for policy makers and future studies. 

Page 19 line 392

“Cognitive impairment, functional decline, and frailty were significantly more prevalent in the poor elderly group, of which the observed inequality was relatively larger among those with cognitive impairment. Employment status and health insurance schemes can be treated as potentially modifiable non-need factors that require attention from health policymakers. Distributions of employment status provided uniform explanation for the widening inequality of all GS phenotypes. Therefore, based on this finding, the study emphasizes the importance of working opportunity in modifying inequitable GS in old age. Bidirectional contributions from inequitable health insurance schemes suggest that equalizing access or consolidating health insurance schemes could be a more promising strategy for the country’s universal health coverage than providing access to any one of the schemes. Inequitable cognitive impairment, employment, and access to insurance schemes required further attention from policymakers to establish a healthy aging society. Nevertheless, future study should evaluate the impact of these socioeconomic contributions overtime and investigate the effectiveness of public health interventions which have been implemented.”

Reviewer 2

1.The title and the introduction of the manuscript sound the social science but the methodology showed the high level of statistical idea. It will be better to explain the idea of index and the statistical methods that show clearer idea of GS and inequlity. 

Response:We thank you for acknowledging the complexity of the concentration index in showing the inequality for audience in general. Thus, we added a clarification into our detailed explanations in our ‘Inequality measurements’ subsection of the method section.

Page 8 line 177

“The index typically ranges from -1 to 1, with a positive value indicating a concentration of the outcome among groups with higher SES (the outcome accumulating more among well-off individuals); a negative value suggesting that individuals in the lower SES group are more likely to experience the outcome than the affluent; and a value of zero indicating no socioeconomic-related inequality.”

2.Table 2 presented the symbol "*" to illustrate the level of p-value significant. I recommend to present with p-value.

Response: We have changed the symbols to p-values in Table2. 

3.For Table 3 and the text explanation, this section was important for answer the research question. However, it hard to understand for less knowledge of statistics. It will be better to explain more for application of this results.

Response: We thank you for giving us the notice and see this one important for readers in general. Thus, we added a sentence explaining the meaning of the decomposition analysis, and added one point in the discussion to indicate that policy makers could also use these decompositions to prioritize the policy to manage these relevant factors.

Page 13 Line 277

“Our decomposition analysis produced AC and %CC which represent the extent of inequality in each variable that was responsible for each GS phenotype.”

4.For the limitation, from "Thus, the current findings may not represent all GS phenotypes..." What types of phenotypes that we can generalized. 

Response: We assessed that our results represent situations of inequality only in the scope of problems: neurocognitive dysfunction, functional impairment, and frailty. Thus, we added a statement to clarify the limitation in generalizability for our study results. 

Page 19 line 387

“Thus, the current findings illustrate situations of inequality only in NCD, FI, and frailty, and might not be generalized to all GS phenotypes, such as decubitus ulcers and polypharmacy.”

---

## [Decision Letter · Decision Letter 1]

15 Aug 2024

PONE-D-23-15911R1Socioeconomic inequalities associated with Geriatric syndrome in Thailand: the results of Fifth National Health Examination SurveyPLOS ONE

Dear Dr. Assanangkornchai,

Thank you for submitting your manuscript to PLOS ONE. After careful consideration, we feel that it has merit but does not fully meet PLOS ONE’s publication criteria as it currently stands. Therefore, we invite you to submit a revised version of the manuscript that addresses the points raised during the review process.

We look forward to receiving your revised manuscript.

Kind regards,

Isaac Akintoyese Oyekola

Academic Editor

PLOS ONE

Journal Requirements:

Additional Editor Comments:

The manuscript examines the socioeconomic inequalities associated with geriatric syndrome in Thailand using the NHES-5. Authors are to attend to the following minor corrections before publication acceptance.

• Generally, research reports are often differentiated from research articles. Hence, authors are advised to take cognizance of that. As it is, the manuscript looks more like research report instead of research article.

• Age of participants (60+ years) should be indicated in Abstract, as mentioned in Methodology “total of 7,365 participants aged 60 years old or older were…”. That is, “Nationally representative data of 7,365 individuals aged 60 years AND ABOVE from the 5th National Health…”.

• Exclusion of participants with missing wealth index data from the NHES-5 should be clearly stated to avoid confusion. Suggestively, in Abstract, “The wealth index was used as the main indicator of socioeconomic status, and PARTICIPANTS with missing wealth index data were excluded.” Also in methodology, “Those (1904 PARTICIPANTS IN TOTAL) who did not report household assets (i.e., missing wealth index values) were excluded from the analysis”.

• Correct minor grammatical errors such as “Therefore, BASED on this finding, the study emphasizes the importance.” Also, “Inequitable cognitive impairment, employment, and access to insurance schemes REQUIRED further attention from policymakers”. Furthermore, “Our findings that the socioeconomic inequalities of NCD, FI, and frailty in the Thai population were also explained by age, education, and working status, while age was considered an actual biological determinant of GS, and that education and employment can be treated as potentially unfairly distributed factors contribute to the literature.”

• “were living with their couples” should be, ‘were living with their SPOUSES’.

• "…not only with the increased likelihood of poorer health outcomes and mortality in the elderly,35–40 but with an increased… NOT ONLY…, BUT ALSO…

• Earlier reviewers noted the need to explain gap in the literature. However, authors addressed this without making reference to previous studies. For instance, “Earlier studies have shown the degree of socioeconomic inequality regarding aging and the general health of older individuals for past decades…” (REQUIRE CITATIONS)

• “Age and work status were significant factors associated with GS, with a higher prevalence among older age groups” (, WIDOWED AND UNEMPLOYED).

• “A higher prevalence of GS was found among individuals with lower socioeconomic status, falling within the first and second terciles of SES.” This is not totally true when compared to 80+ elderly, widowed, residents in the south and unemployed. Hence, the word ‘higher’ should be reconsidered. Generally, while the reason for not speaking to geographical distribution of GS is best known to the authors, it is important to retain the distribution accordingly: older elderly (80+ years), widowed, southern residence, unemployed, poorest, 70-79 years elderly, females, and northern residence. Afterward, Northeastern residence, average SES, lower education, single, having UCS, having CSMBS, and separated come next in the distribution of GS. By cognitive impairment, older elderly (80+ years), single, as well as widowed and poorest. By functional impairment, older elderly (80+ years), southern residence, and widowed. By frailty, older elderly (80+ years), unemployed, and widowed.

• “…of which the observed inequality was relatively larger among those with cognitive impairment.” EVEN MUCH LARGER AMONG FUNCTIONALLY IMPAIRED.

• “We used age, gender, marital status, education, working status, health insurance, and region of residence, which were investigated in the NHES-5.28” (Aekplakorn W, Chariyalertsak S, Kessomboon P, Assanangkornchai S, Taneepanichskul S, Putwatana P. Prevalence of diabetes and relationship with socioeconomic status in the Thai population: National health examination survey, 2004–2014. J. Diabetes Res. 2018 ;2018:e1654530.) THE REFERENCE DOES NOT SEEM TO MATCH/PROVE THE CLAIM.

• List of references seem to be unnecessarily too lengthy, and missing doi. should be included.

Reviewers' comments:

Reviewer's Responses to Questions

**Comments to the Author**

1. If the authors have adequately addressed your comments raised in a previous round of review and you feel that this manuscript is now acceptable for publication, you may indicate that here to bypass the “Comments to the Author” section, enter your conflict of interest statement in the “Confidential to Editor” section, and submit your "Accept" recommendation.

Reviewer #2: All comments have been addressed

2. Is the manuscript technically sound, and do the data support the conclusions?

Reviewer #2: Yes

3. Has the statistical analysis been performed appropriately and rigorously? 

Reviewer #2: Yes

4. Have the authors made all data underlying the findings in their manuscript fully available?

Reviewer #2: Yes

5. Is the manuscript presented in an intelligible fashion and written in standard English?

Reviewer #2: Yes

6. Review Comments to the Author

Reviewer #2: The authors showed the statistical methods to the national survey that can answer the research question.

7. PLOS authors have the option to publish the peer review history of their article (what does this mean?). If published, this will include your full peer review and any attached files.

Reviewer #2: No

---

## [Author Response · Author response to Decision Letter 1]

19 Sep 2024

General response

First, we would like to express our thanks for your thorough editing and valuable suggestions for improving the manuscript. We have carefully reviewed and addressed each of your comments point-by-point.

Point-by-point response to the editor’s comments

• Generally, research reports are often differentiated from research articles. Hence, authors are advised to take cognizance of that. As it is, the manuscript looks more like research report instead of research article.

Response: We acknowledge the merits of the study and would like to thank you for your constructive comments.

• Age of participants (60+ years) should be indicated in Abstract, as mentioned in Methodology “total of 7,365 participants aged 60 years old or older were…”. That is, “Nationally representative data of 7,365 individuals aged 60 years AND ABOVE from the 5th National Health…”.

Response: We thank you for highlighting the need for clarifications, and we have revised the relevant text per your suggestions. 

Revisions have been made on page 2, line 31

• Exclusion of participants with missing wealth index data from the NHES-5 should be clearly stated to avoid confusion. Suggestively, in Abstract, “The wealth index was used as the main indicator of socioeconomic status, and PARTICIPANTS with missing wealth index data were excluded.” Also in methodology, “Those (1904 PARTICIPANTS IN TOTAL) who did not report household assets (i.e., missing wealth index values) were excluded from the analysis”.

Response: We thank you for highlighting the need for clarifications, and we have revised the relevant text per your suggestions.

Revisions have been made on page 2 line 35 and page 5 line 120

• Correct minor grammatical errors such as “Therefore, BASED on this finding, the study emphasizes the importance.” Also, “Inequitable cognitive impairment, employment, and access to insurance schemes REQUIRED further attention from policymakers”. Furthermore, “Our findings that the socioeconomic inequalities of NCD, FI, and frailty in the Thai population were also explained by age, education, and working status, while age was considered an actual biological determinant of GS, and that education and employment can be treated as potentially unfairly distributed factors contribute to the literature.”

Response: Thank you for highlighting the need for revisions. We have updated the relevant text according to your suggestions

Revisions have been made on:

Page 17 line 326-330 (the whole sentence is rearranged)

Page 20 line 399 (“based on this finding,” has been removed) 

• “were living with their couples” should be, ‘were living with their SPOUSES’.

Response: We thank you for highlighting the need for revisions, and we have revised the relevant text per your suggestions.

Revisions have been made on page 10 line 233

• "…not only with the increased likelihood of poorer health outcomes and mortality in the elderly,35–40 but with an increased… NOT ONLY…, BUT ALSO…

Response: Thank you for highlighting the need for revisions. We have updated the relevant text according to your suggestions

Revisions have been made on page 16 line 320

• Earlier reviewers noted the need to explain gap in the literature. However, authors addressed this without making reference to previous studies. For instance, “Earlier studies have shown the degree of socioeconomic inequality regarding aging and the general health of older individuals for past decades…” (REQUIRE CITATIONS)

Response: We thank you for your comment. we have revised our statement to more precise (using “in the last decade” instead of “for the past decades”) and added the relevant reference.

Revisions have been made on page 5 line 99

• “Age and work status were significant factors associated with GS, with a higher prevalence among older age groups” (, WIDOWED AND UNEMPLOYED).

Response: Thank you for highlighting the key points to emphasize in the results section. We have revised the relevant text in accordance to your suggestions.

Revisions have been made on page 14 line 269 – 271

• “A higher prevalence of GS was found among individuals with lower socioeconomic status, falling within the first and second terciles of SES.” 

This is not totally true when compared to 80+ elderly, widowed, residents in the south and unemployed. Hence, the word ‘higher’ should be reconsidered. Generally, while the reason for not speaking to geographical distribution of GS is best known to the authors, it is important to retain the distribution accordingly: older elderly (80+ years), widowed, southern residence, unemployed, poorest, 70-79 years elderly, females, and northern residence. Afterward, Northeastern residence, average SES, lower education, single, having UCS, having CSMBS, and separated come next in the distribution of GS. By cognitive impairment, older elderly (80+ years), single, as well as widowed and poorest. By functional impairment, older elderly (80+ years), southern residence, and widowed. By frailty, older elderly (80+ years), unemployed, and widowed.

Response: We appreciate your constructive comments and suggestions regarding our wording. We have carefully revised the relevant text to clearly mention the high prevalence of geriatric syndrome features in relation to the lower socioeconomic status terciles only.

Our revised text is: “In regard to SES, a higher prevalence of GS was found among individuals with lower socioeconomic status, falling within the first and second terciles of SES,” on page 16, lines 308–309.

• “…of which the observed inequality was relatively larger among those with cognitive impairment.” EVEN MUCH LARGER AMONG FUNCTIONALLY IMPAIRED.

Response: Thank you for your comment. In response, we carefully reviewed our conclusion and revised the relevant text for greater clarity. The larger magnitude refers to the concentration index of cognitive impairment (CI) (C = -0.220) compared to the other two phenotypes: functional impairment (C = -0.082) and frailty (C = -0.130). To clarify, we have added the term 'socioeconomic' to emphasize that the greater magnitude of the concentration index reflects a higher level of socioeconomic inequality in CI within our population.

Our revised text is: “Cognitive impairment, functional decline, and frailty were significantly more prevalent among the poor elderly group, with the observed socioeconomic inequality being relatively larger for those with cognitive impairment.” On page 19 line 395.

• “We used age, gender, marital status, education, working status, health insurance, and region of residence, which were investigated in the NHES-5.28” (Aekplakorn W, Chariyalertsak S, Kessomboon P, Assanangkornchai S, Taneepanichskul S, Putwatana P. Prevalence of diabetes and relationship with socioeconomic status in the Thai population: National health examination survey, 2004–2014. J. Diabetes Res. 2018 ;2018:e1654530.) THE REFERENCE DOES NOT SEEM TO MATCH/PROVE THE CLAIM.

Response: We thank you for your comment and understand that the reference title and its main analysis focused on diabetes. However, we selected this article as our reference because it contains the clearest English description (as provided by the principal investigator of the project) of the National Health Survey Method, which was used up until the most recent survey, from which our data was drawn. Additionally, the original report, which includes methodological details, is only available in Thai and may not offer as clear an explanation of the survey methodology for an English-speaking audience as the cited reference.

• List of references seem to be unnecessarily too lengthy, and missing doi. should be included.

Response: We understand that the number of references should be kept optimal. However, after we trimmed out the references in the list, almost all the references have been kept due to their relevance to the current study. Any retracted references can be tracked by track changes in the manuscript file as well as are detailed as follows. The articles’ available dois have also been added. 

1. Removal due to duplication (n = 1)

Ref8: Inouye SK, Studenski S, Tinetti ME, Kuchel GA. Geriatric syndromes: clinical, research and policy implications of a core geriatric concept. J Am Geriatr Soc. 2007;55(5):780-91.

2. Removal due to redundancy (n = 4)

Ref35. Bornæs O, Andersen AL, Houlind MB, Kallemose T, Tavenier J, Aharaz A, et al. Mild cognitive impairment is associated with poorer nutritional status on hospital admission and after discharge in acutely hospitalized older patients. Geriatrics. 2022; 7(5):95.

Ref39. Kojima G, Iliffe S, Jivraj S, Walters K. Association between frailty and quality of life among community-dwelling older people: A systematic review and meta-analysis. J Epidemiol Community Health. 2016 Jul; 70(7):716-21. 

Ref40. Kojima G, Iliffe S, Walters K. Frailty index as a predictor of mortality: A systematic review and meta-analysis. Age Ageing. 2018 ; 47(2):193-200.

Ref59: Blom J, den Elzen W, van Houwelingen AH, Heijmans M, Stijnen T, Van den Hout W, et al. Effectiveness and cost-effectiveness of a proactive, goal-oriented, integrated care model in general practice for older people. A cluster randomised controlled trial: Integrated Systematic Care for older People—the ISCOPE study. Age and Ageing. 2016; 45(1):30-41.

---

## [Editor Report · Decision Letter 2]

24 Sep 2024

Socioeconomic inequalities associated with Geriatric syndrome in Thailand: the results of Fifth National Health Examination Survey

PONE-D-23-15911R2

Dear Dr. Assanangkornchai,

We’re pleased to inform you that your manuscript has been judged scientifically suitable for publication and will be formally accepted for publication once it meets all outstanding technical requirements.

Kind regards,

Isaac Akintoyese Oyekola

Academic Editor

PLOS ONE

Additional Editor Comments (optional):

Having carefully addressed editor’s comments, the manuscript may be considered for publication. However, authors must address the following very minor corrections among others:

• Full stop after “…were decomposed and shown in terms of percentage and direction” in Abstract.

• Change ‘or’ to ‘and’ in “We analyzed the data of individuals aged 60 years or above”.

• Include ‘s’ in acknowledgement – Acknowledgements
---

## [Editor Report · Acceptance letter]

30 Sep 2024

PONE-D-23-15911R2 

PLOS ONE

Dear Dr. Assanangkornchai, 

I'm pleased to inform you that your manuscript has been deemed suitable for publication in PLOS ONE. Congratulations! Your manuscript is now being handed over to our production team.

Kind regards, 

on behalf of

Dr. Isaac Akintoyese Oyekola 

Academic Editor

PLOS ONE